# Restraint of Human Skin Fibroblast Motility, Migration, and Cell Surface Actin Dynamics, by Pannexin 1 and P2X7 Receptor Signaling

**DOI:** 10.3390/ijms22031069

**Published:** 2021-01-22

**Authors:** Carolina Flores-Muñoz, Jaime Maripillán, Jacqueline Vásquez-Navarrete, Joel Novoa-Molina, Ricardo Ceriani, Helmuth A. Sánchez, Ana C. Abbott, Caroline Weinstein-Oppenheimer, Donald I. Brown, Ana María Cárdenas, Isaac E. García, Agustín D. Martínez

**Affiliations:** 1Centro Interdisciplinario de Neurociencia de Valparaíso, Instituto de Neurociencia, Facultad de Ciencias, Universidad de Valparaíso, Valparaíso 2360102, Chile; carolina.flores@cinv.cl (C.F.-M.); jaime.maripillan@cinv.cl (J.M.); jacqueline.vasquez@cinv.cl (J.V.-N.); joel.novoa@cinv.cl (J.N.-M.); ricardo.ceriani@cinv.cl (R.C.); helmuth.sanchez@cinv.cl (H.A.S.); ana.abbott@cinv.cl (A.C.A.); ana.cardenas@uv.cl (A.M.C.); isaac.garcia@uv.cl (I.E.G.); 2Programa de Doctorado en Ciencias, Mención Neurociencia, Universidad de Valparaíso, Valparaíso 2340000, Chile; 3Escuela de Química y Farmacia, Facultad de Farmacia, Universidad de Valparaíso, Valparaíso 2360102, Chile; caroline.weinstein@uv.cl; 4Centro de Investigación Farmacopea Chilena, Valparaíso 2360102, Chile; 5Laboratorio de Biología de la Reproducción y del Desarrollo, Instituto de Biología, Facultad de Ciencias, Universidad de Valparaíso, Valparaíso 2340000, Chile; donald.brown@uv.cl; 6Laboratorio de Fisiología Molecular y Biofísica, Facultad de Odontología, Universidad de Valparaíso, Valparaíso 2360004, Chile

**Keywords:** pannexin 1, actin cytoskeleton, cell migration, wound healing, human dermal fibroblasts, purinergic receptor

## Abstract

Wound healing is a dynamic process required to maintain skin integrity and which relies on the precise migration of different cell types. A key molecule that regulates this process is ATP. However, the mechanisms involved in extracellular ATP management are poorly understood, particularly in the human dermis. Here, we explore the role, in human fibroblast migration during wound healing, of Pannexin 1 channels and their relationship with purinergic signals and in vivo cell surface filamentous actin dynamics. Using siRNA against Panx isoforms and different Panx1 channel inhibitors, we demonstrate in cultured human dermal fibroblasts that the absence or inhibition of Panx1 channels accelerates cell migration, increases single-cell motility, and promotes actin redistribution. These changes occur through a mechanism that involves the release of ATP to the extracellular space through a Panx1-dependent mechanism and the activation of the purinergic receptor P2X7. Together, these findings point to a pivotal role of Panx1 channels in skin fibroblast migration and suggest that these channels could be a useful pharmacological target to promote damaged skin healing.

## 1. Introduction

Pannexin (Panx) channels are transmembrane proteins involved in cell communication. There are three Panx isoforms: Panx1, Panx2, and Panx3, which form channels with a large vestibule and increase their activity under several physiological stimuli [1,2,3,4]. It is well documented that the activity of Panx1 channels allows adenosine triphosphate (ATP) release to the extracellular space [1], which activates purinergic receptors and causes the activation of intracellular signaling cascades [5]. In addition, it has been suggested that Panx1 channels interact with the ionotropic purinergic receptor, P2X7 (P2X7R) [1,6,7], such that stimulation of P2X7R with ATP increases the open probability of Panx1 channels [8], suggesting the existence of a positive feedback mechanism that results in the release of additional ATP. It has been proposed that ATP-mediated purinergic signaling plays important and diverse roles in skin physiology and wound healing [9]. For example, the activation of the inflammatory response orchestrates a fine-tuning mechanism required to achieve effective cellular migration, which involves rearrangements of the cytoskeletal dynamics [10,11] mediated by actin and the small Rho GTPases family [12].

Despite the proposed role of connexins during normal skin repair and wound healing [13,14,15,16,17,18], little is known about the contribution of Panxs to these processes. It is well known that Panx1 and Panx3 are expressed in human and rodent keratinocytes [19,20,21,22,23,24], whereas Panx2 is expressed in the central nervous system [25,26]. Interestingly, the overexpression of Panx1 in rat epidermal keratinocytes reduces their proliferation rate and disrupts normal differentiation [20]. In contrast, the absence of Panx1 promotes fibroblast proliferation and reduces their differentiation in response to TGF-β [21]. Moreover, skin Panx1 expression, which diminishes with age, is up-regulated after skin injury [20], suggesting a role in the wound healing process. Penuela and colleagues [21] found that keratinocytes from Panx1 knock-out animals (Panx1 KO) migrate more efficiently than do wild-type (WT) keratinocytes, suggesting that Panx1 is involved in regulating the migration rate in this cell type in response to tissue damage. In addition, in dendritic cells, Panx1 participates in a positive feedback loop that potentiates dendritic cell motility induced by ATP that permeates through these channels [27]. By contrast, Wicki-Stordeur and Swayne [28], using small interfering RNA (siRNA) against Panx1, found that reduction in Panx1 expression slows down cell migration in neuroblasts. Overall, these results suggest that Panx1 is involved in cell motility and migration and, depending on cell type, the activity of Panx1 channels can accelerate or reduce cell migration. However, whether these processes are under Panx regulation in human skin cells has yet to be determined. Here, we investigated the possible role of Panx1 channels and purinergic signaling in human dermal fibroblast (HDFs) migration and proposed a possible mechanism for wound repair that involves modulation of cell surface F-actin dynamics.

## 2. Results

### 2.1. Inhibition or Blockage of Panx1 Channels Accelerates the In Vitro Wound Healing of HDFs

As previously described and characterized by [21,23], we found that using RT-PCR and Western blot analyses of HDF cultures, Panx1 and Panx3 are expressed in primary HDFs (Figure 1A,B). We then determined the role of Panx1 channels during the migration process using an in vitro wound-healing assay on cultured HDFs (see Section 4). Under control conditions, HDFs repopulated about 50% of the wounded area 12 h post-injury, reaching almost complete closure of the wound area after 24 h post-injury (Figure 1C,D; Appendix A). To test if Panxs participate in this process, HDFs were bathed in a solution containing either 200 µM probenecid (PBN), a non-selective Panx1 channel inhibitor, or 100 µM ^10^Panx, a selective Panx1 blocker [29,30]. Both treatments accelerated the healing process, leading to nearly complete wound closure within 12 h post-injury. Importantly, this effect was not observed when cells were treated with a scrambled mimetic peptide (Sc ^10^Panx, 100 µM) (Figure 1C,D).

To more specifically address the potential role of Panx1 or Panx3 in wound closure, we used siRNA-GFP for selective reduction in Panx expression in HDFs. The expression levels of Panx1 and Panx3 were significantly reduced by around 70% and 60%, respectively, after 24 h of transfection with the specific plasmid siRNA compared with the control plasmid (Figure 2C). We then evaluated the migration index, defined as the number of cells expressing siRNA-GFP in the migration front of the wound area (10–15 cell rows from the migration front; see Section 4), compared to non-transfected cells. As expected, the proportion of HDF cells expressing the control plasmid did not change (Figure 2A,B; Appendix A) between 0 and 9 h after the wound. By contrast, the proportion of cells expressing the siRNA against Panx1 increased in a time-dependent manner with respect to non-transfected cells in the migration front (Figure 2A,B; Appendix A). A similar experimental strategy was applied to investigate the role of Panx3 in the healing process. Using a siRNA against Panx3, we observed that the migration index for cells expressing siRNA-Panx3 reached a maximum level at 3 h post-injury, with no further increases during the remaining recording time (Figure 2A,B; Appendix A).

To further address the role of Panx1 in wound healing, we used primary fibroblast cultures from the skin of Panx1 knock-out mouse (MDF^Panx1−/−^), which express Panx3 normally (Appendix A). The wound-healing assay in MDF^Panx1−/−^ revealed a tendency to accelerate wound closure within the first 12 h when compared with HDF and WT mouse dermal fibroblast (MDF) cultures (Appendix A).

### 2.2. Activity of Panx1 Channels during Wound Healing in HDF Cultures

We have determined that Panx1 channels are involved in the regulation of the migration rate of HDFs. To explore to what extent the functional activity of Panx1 channels is affected during wound healing, we determined the temporal course of ethidium uptake (Etd) [31] in control cells and in cells treated with PBN, ^10^Panx, or Sc ^10^Panx (Figure 3A; Appendix A). An increased number of Etd-stained nuclei was found 3 h after scratching in CTL and Sc ^10^Panx-treated HDFs, which was greatly reduced in cells treated with the Panx1 channel blockers PBN and ^10^Panx (Figure 3A).

In order to better quantify the Panx1 channel’s activity, we determined the Etd uptake rate at different times during wound healing. The relative fluorescent intensities measured at the edge of the wound area immediately after inducing the wound (0 h) showed a low basal Etd uptake rate (0.83 ± 0.03 F/F_0_ min^−1^), which was not affected by Panx1 blockers (0.53 ± 0.01 F/F_0_ min^−1^ for PBN; 0.65± 0.01 F/F_0_ min^−1^ for ^10^Panx) (Figure 3B). However, an increased Etd uptake was observed at 3, 6, and 9 h, after inducing the wound (3.03 ± 0.05 F/F_0_ min^−1^ at 3 h; 4.52 ± 0.05 F/F_0_ min^−1^ at 6 h; 5.36 ± 0.03 F/F_0_ min^−1^ at 9 h), which was sensitive to Panx1 blockers [(at 3 h (0.86 ± 0.02 F/F_0_ min^−1^ for PBN; 0.71 ± 0.04 F/F_0_ min^−1^ for ^10^Panx); at 6 h (0.72 ± 0.05 F/F_0_ min^−1^ for PBN; 0.62 ± 0.05 F/F_0_ min^−1^ for ^10^Panx); at 9 h (0.38 ± 0.04 F/F_0_ min^−1^ for PBN; 0.49 ± 0.05 F/F_0_ min^−1^ for ^10^Panx)] (Figure 3B). Treatment with Sc ^10^Panx did not significantly affect dye uptake [at 0 h (0.58 ± 0.04 F/F_0_ min^−1^); at 3 h (3.02 ± 0.03 F/F_0_ min^−1^), at 6 h (4.34 ± 0.01 F/F_0_ min^−1^); at 9 h (5.24 ± 0.05 F/F_0_ min^−1^)] (Figure 3B). These results reveal that Panx1 channels are active during wound healing assays of HDF cultures and that this channel’s activity increases as wound closure progresses.

It has been widely reported that ATP can be released through Panxs channels [1,30]. Thus, we determined the amount of ATP released by HDFs during wound closure using a luminescence-based assay. The increase in extracellular ATP concentration induced by the wound was significantly reduced by 200 µM PBN (Figure 3C). In addition, we measured extracellular ATP concentration after 12 h of wound healing in MDF and MDF^Panx1−/−-^ cultures and found a significant increase in ATP in MDFs compared with MDF^Panx1−/−^; this increase was prevented by the Panx1 blocker PBN (200 µM PBN) (Appendix A). These results suggest that Panx1 channels contribute to the wound healing-induced ATP release from these cells. 

As mentioned above, cell proliferation is also part of the mechanism triggered by wound healing. To rule out the possibility that wound closure following Panx1 channels inhibition was due to cell proliferation stimulation, we used a proliferating cell nuclear antigen (PCNA) antibody, a well-known cell proliferation marker [28]. We did not observe significant differences in the expression of PCNA during wound closure at any time and under any treatment tested (Appendix A), which strongly suggests that the blockade of Panx1 channels modulates wound closure of HDF cultures by a mechanism that involves cell migration rather than cell proliferation.

### 2.3. Blocking the Purinergic P2X7 Receptor Significantly Accelerates Wound Healing in HDF Cultures

Since we showed that ATP release by HDFs is strongly reduced by blockade of Panx1 channels, we decided to evaluate if extracellular ATP modulates in vitro healing in HDFs. For this, we used apyrase, which catalyzes the degradation of extracellular ATP. Cells that were subjected to the wound assay were treated with 10 U/mL of apyrase for 24 h. We found that apyrase treatment accelerated the wound area’s reduction (Figure 4A,B), suggesting that endogenous ATP slows down migration.

It is known that HDF cells express purinergic receptors [32,33]. Thus, we sought to investigate if there is crosstalk between Panx1 channels and purinergic receptors during the wound healing process. We confirmed the P2X7 and P2Y2 receptors (P2X7R and P2Y2R) expression in these cells (Figure 4C). Then, we evaluated if these receptors are involved in the wound healing process by performing the wound healing assay in the presence of purinergic antagonists including 100 µM suramin (for P2X and P2Y), 200 µM brilliant blue G (BBG, for P2X), or 10 µM A-740003 (for P2X7R) [34]. All these antagonists were found to accelerate the wound healing process and to reduce the wound area, with A-740003 being the most effective (Figure 4D,E). Indeed, this antagonist reduced the wound area by almost 90% 12 h after the injury (scrape) (Figure 4D,E), similar to the effect obtained using Panx1 blockers (Figure 1). Interestingly, the migration of HDF cells was not affected by treatment with a selective agonist or antagonist of P2Y2R, 10 µM MRS-2768, and 10 µM AR-C118925 (Appendix A). These results support the involvement of P2X7Rs in the healing process.

To determine if there is crosstalk between P2X7R and Panx1 channels during wound healing, we applied the P2X7R antagonists in fibroblasts derived from MDF^Panx1−/−^ mice. In contrast to the effect observed in HDFs, the purinergic blockers did not decrease the wound area in MDF^Panx1−/−^, and suramin and BBG increased it (Appendix A), probably unmasking the effects of other purinergic receptors on cell migration and/or proliferation [9,35,36]. Taken together, the pharmacological inhibition and genetic knock-out data strongly suggest that P2X7R contributes to cell migration through a process that depends on Panx1 channels.

To determine if extracellular ATP is mediating this process, we added 1 mM ATP to the extracellular milieu and measured the wound healing rate (Figure 4F,G). We found that exogenous ATP slightly accelerated (~15%) the wound healing process. Nevertheless, incubation with 1 mM ATP did reduce the healing process acceleration induced by PBN and A-740003, suggesting that extracellular ATP is acting downstream of Panx1 channels. Since it seems contradictory to the observation that 1 mM ATP tends to accelerate the healing process in control HDFs, but the same treatment reduced the closure area induced by treatments with PBN or A-740003 (Figure 4F,G), we investigated whether the effect of ATP on the wound healing process is concentration-dependent. Indeed, incubation of HDF cells with different ATP concentrations clearly showed that only higher ATP doses produce significant acceleration of wound healing in HDF cultures (Appendix A).

### 2.4. Blocking Panx1 Channels Increases Cell Motility and Triggers Fast Rearrangements of Cell Surface Actin Filaments

As seen in Figure 1, the rapid wound healing elicited in vitro by Panx1 channel blockers strongly suggests that cell motility is somehow under Panx1 control. To test this hypothesis, we measured rapid cell shape changes over time and determined the motility index (see Section 4). While the motility index did not change under control conditions, we found that the application of 200 µM PBN rapidly increased the motility index (Figure 5A,B), which was associated with an increase in the displacement of isolated HDFs (Figure 5C).

To further study how the Panx1 channel activity impacts the cellular machinery required for cellular motion, we investigated whether the blockade of these channels affects cell surface actin filaments. For this, HDFs were transfected with LifeAct-RFP, a fungus-derived peptide that binds to F-actin, allowing real-time visualization of microfilament dynamics [37]. Performing TIRF microscopy and video imaging, we observed F-actin stress fibers’ depolymerization when the cells were treated with PBN for 70 min (Figure 5D,E; 120 min of recording time), and this depolymerization was further increased at 180 min (Figure 5D,E). Together, these results suggest that signaling through Panx1 channels controls actin microfilament dynamics.

## 3. Discussion

Wound healing is a complex process that is essential for skin homeostasis. Dermal fibroblasts are essential for cutaneous wound repair because they migrate to the side of the damage, repopulate the wound area, and remodel fibrin and collagen deposits. All these processes require several finely orchestrated steps [38].

The present study shows that Panx1 channels in conjunction with P2X7Rs regulate cell migration during the wound healing process. Our findings using HDFs are consistent with those published with mice keratinocytes, in which cells from Panx1 knock-out animals (Panx1 KO) were found to migrate more efficiently than did wild-type (WT) cells [21]. However, the findings that inhibition of Panx1 channels with carbenoxolone or ^10^Panx prevents the migration of neutrophils induced by chemoattractants [39,40,41] and that Panx1 knockdown with siRNA or its blockage with PBN reduces the rate of wound closure and enhances neurite outgrowth in N2a cells [28,42] suggest that the role of Panx1 channels and of purinergic signaling in cell migration is cell type-dependent. We cannot discount the possibility that Panx3 is also involved in regulating cell migration since treatment with siRNA against Panx3 also accelerates the wound healing process of HDF cultures, although it did so to a lesser extent than did inhibition of Panx1. In addition, recent findings point to a role of Panx3 in in vivo wound healing after skin punch lesions, and Panx3-knock-out mice have reduced healing and collagen remodeling compared to wild-type animals [22].

Panx1 channels have been reported to regulate the proliferation of human subcutaneous fibroblasts, glioma cell lines, neural stem cells, and progenitor cells [20,23,28,43,44]. Therefore, we performed the wound healing experiments in HDF cells synchronized by serum starvation before and during the wound healing protocol. Serum deprivation can induce G0 phase cell cycle arrest and reduce the proliferation of most cell types [45]. Accordingly, our PCNA immunoreactivity measurements showed that Panx1 channel blockers do not affect cell proliferation, confirming that the healing process occurred through a process of migration rather than through cell proliferation.

It has previously been shown that scratch wounds cause the release of approximately ~50–100 µM ATP from damaged cells to the culture medium [46]. Under basal conditions, ATP is present in concentrations of approximately 1 mM inside the cell and approximately 1 nM–1 µM outside the cell [47,48]. In our experiments, we observed ATP media concentrations in a nanomolar range after inducing the wound. The finding that Etd uptake rate and ATP release were largely reduced by treatment with Panx1 blockers suggests that Panx1 channels are open during the wound healing process and contribute to a signaling mechanism involved in the activation and migration of HDFs.

Since treatment with apyrase also tends to accelerate HDF migration during wound healing, we thought that purinergic signaling is also involved in controlling HDF migration associated with the activation of Panx1 channels. ATP released through Panx1 channels activates purinergic receptors, particularly P2X7Rs [1,7,49]. Consistent with this hypothesis, treatments with A-740003 or BBG, two P2X7Rs antagonists, also accelerate the wound healing process, suggesting that Panx1 channels and P2X7Rs modulate HDF migration during wound healing through an autocrine/paracrine signaling process. In fact, our findings that MDF^panx1−/−^ migrates faster than wild-type MDFs, and even faster than HDFs, may be related to the reduced ATP concentration found in the MDF^panx1−/−^ extracellular medium during the process of wound closure. The finding that inhibition of Panx1 channels with PBN causes a significant reduction in ATP release from HDFs and MDFs, but not from MDF^panx1−/−^ (Appendix A), during wound healing strongly suggests that Panx1 channels are the major contributor to ATP release in fibroblasts. Additionally, we demonstrated that the acceleration of the HDF migration rate treated with Panx1 blockers is partially reverted by exogenous ATP, supporting the idea that activation of purinergic receptors is downstream of Panx1 channels. Another possible mechanism may involve forming a functional intermolecular complex between Panx1 channels and P2X7Rs [30]. The latter is consistent with our finding that the P2X7Rs blocker A-740003 is capable of inhibiting ATP release from HDFs and MDFs, but not from MDF^panx1−/−^ (Appendix A), suggesting that in our model, ATP may also activate Panx1 channels, supporting the idea that the activation of P2X7R could be both upstream and downstream of Panx1 channel activation. The latter agrees with previous findings that P2X7R activation causes ATP release [50,51] and activation of Panx1 channels [7,8,52]. However, we cannot discard that the inhibition of P2X7R with A-740003 affects the Panx1-independent mechanism of ATP release. For example, it has been proposed that P2X7R itself can form a non-selective plasma membrane channel (or macropore) that is permeable to large hydrophilic molecules, such as dyes or ATP [53,54].

On the other hand, it has been described that the elevation of extracellular ATP leads to Panx1 internalization, reducing Panx1 surface expression, and these processes require activation of P2X7Rs [55,56]. Therefore, we cannot discard a mechanism in which the over-stimulation of P2X7R with a high ATP concentration may accelerate wound healing. Consistent with this hypothesis, we found that incubation with a high ATP concentration accelerated HDF migration during wound healing. Similarly, exogenous ATP accelerates dendritic cell migration during injury by a mechanism that requires Panx1 channels and P2X7Rs [27].

We cannot dismiss the possibility that other purinergic receptors besides P2X7Rs may be involved in HDF migration regulation. It has been shown that ATP released through Panx1 activates P2Y2Rs, which then activate neutrophils’ migration [39,40,41]. In corneal injury, downregulation of P2Y2Rs decreases the rate of migration of epithelial cells [57]. However, although we found that HDFs express P2Y2Rs, treatment with specific P2Y2R antagonists did not change the wound healing rate, indicating that in HDFs, P2YRs were not involved in regulating migration during wound healing. Extracellular ATP can be dephosphorylated to adenosine by the coordinated action of ecto-apyrase and ecto-5′-nucleotidase, and adenosine induces loss of actin stress fibers and cell contraction [58,59] processes that are both required for migration. Unfortunately, we could not test adenosine receptor antagonists or other purinergic receptor antagonists in this study; hence, future studies will be necessary to carry out a more complete pharmacologic analysis of the purinergic signaling involved in this process.

The mechanism by which Panx1 channels control cell motility remains unclear. Still, it is logical to look at the cytoskeleton since it is well known that actin-based cell migration is a key process for morphogenesis and wound healing [60]. Myosin-induced contraction and disassembly of the actin networks generate contraction and forward translocation of the cell body, which is the basis for all cell migration processes [61]. In this regard, an interaction between Panx1 and the actin-related protein 3 (Arp3) [28], an actin cytoskeleton-modulating protein [62], has been demonstrated. Accordingly, pretreatment of neutrophils with carbenoxolone and ^10^Panx prevented actin polymerization and cell polarization [41]. Moreover, we found that inhibition of Panx1 channels accelerates actin dynamics (depolymerization) on the cell surface, increasing the cell motility index. On the other hand, it has been described that extracellular ATP in the micromolar to millimolar range markedly redistributes actin filaments towards the plasma membrane [63,64]. The nature of the distribution of actin filaments varies depending on cell types, such as fibroblast, and the exposure to high concentrations of extracellular ATP, above or equal to 1 mM, is known to increase contractile stress fibers [65,66] and can generate contractile forces by depolymerization of cytoskeletal filaments [67]. Yet higher concentrations of extracellular ATP have an inhibitory effect on actin nucleation/polymerization (e.g., by phosphorylating a capping or actin-binding protein) [68,69].

On the other hand, Bao and co-workers [70] proposed that ATP released through Panx1 channels triggers a signaling cascade that involves increases in intracellular Ca^2+^ that regulate actomyosin-mediated mechanics during cell migration. However, this hypothesis needs to be tested. Another more speculative hypothesis is that Panx1 channels, by allowing constitutive leakage of intracellular ATP to the extracellular milieu or increasing influx of extracellular Ca^2^, control the intracellular Ca^2+^ and ATP concentrations, both of which are critical for actin polymerization. Clearly, the mechanism that links Panx1 channels activity and cytoskeletal dynamics during migration need to be further explored.

In summary, we found that Panx1 channels and purinergic signaling through P2X7Rs and/or other yet to be identified purinergic receptors modulate HDF migration. Figure 6 shows the simplest model that can explain our results, in which the activity of Panx1 channels favors ATP release after wound induction, and ATP, acting through P2X7Rs, reduces fibroblast migration. Thus, blocking Panx1 channels or P2X7Rs accelerates migrations during in vitro wound healing. However, higher ATP doses, acting through P2X7R or other purinergic receptors, accelerate cell motility and cell migration, as in the find-me signals.

## 4. Materials and Methods

### 4.1. Animals

Wild-type C57BL/6J or C57BL/6J-Panx1^-/-^ mice were used. Panx1^-/-^-KO mice were generated as described previously [71]. Mice were housed at 22 °C under constant humidity (55%), with a 12/12 h dark–light cycle, with free access to food and water. The use and animal care protocols were approved by the Ethics and Animal Care Committee of Universidad de Valparaíso for doctoral project (BEA-139; 10 June 2019) and for FONDECYT grant 1171240 (BEA-102; 22 May 2017).

### 4.2. Human and Mice Fibroblast Cell Cultures

Primary human dermal fibroblasts (HDFs) were provided by Dr. Caroline Weinstein. HDFs were obtained from biopsies of healthy abdominal skin from 4 patients who underwent regenerative autograft surgery. Only remnant, clinically healthy skin, not required for diagnostic or surgery purposes, was used. HDF cultures from biopsies were prepared as in [72] with modification. First, the biopsy was extensively washed with sterile phosphate saline buffer (pH 7.35). Then, it was mechanically disrupted, using sterile scalpels, and incubated in collagenase (3 mg/mL) (Nordmark, Uetersen, GE; medical grade) for 1 h within an incubator (37 °C, 5% CO_2_, and humidified atmosphere). After this, the mixture was centrifuged at 553× g, and the pellet was cultured in Dulbecco’s modified Eagle’s medium: nutrient mixture F-12 (Thermo Fisher Scientific, Cambridge, MA, USA), 10% inactivated fetal bovine serum (Biological Industries, Cromwell, CT, USA) 10,000 IU/mL penicillin, and 10 μg/mL streptomycin (Thermo Fisher Scientific, Waltham, MA, USA), in a 5% CO_2_ atmosphere at 37 °C. HDFs from abdominal skin were characterized mainly by morphological criteria. These cultures present long spindle-shaped cells, stellate-shaped cells with fibroblastic extensions, and small round dividing primary cells.

Primary cell cultures of mice dermal fibroblasts (MDFs) were obtained from isolated neonatal skin cells essentially as described previously [73]. Briefly, the skin of euthanized 0–4-day-old mice was incubated overnight with a solution containing 3 mg/mL collagenase II (300 U/mg, Worthington, Lakewood, NJ, USA), 100 IU/mL penicillin, and 100 µg/mL streptomycin (HyClone, Missouri, TX, USA). The epidermis was separated from the dermis, minced, and centrifuged (at 14,000 rpm for 5 min at 20 °C). Cells were grown until confluence in culture flasks containing Dulbecco’s modified Eagle’s medium: nutrient mixture F-12 (Thermo Fisher Scientific, Waltham, MA, USA), 10% heated inactivated fetal bovine serum (Gibco, Waltham, MA, USA), and 10,000 IU/mL penicillin/streptomycin (Thermo Fisher Scientific, Waltham, MA, USA), in a 5% CO_2_ atmosphere and maintained at 37 °C. The maximum number of passages for all primary cultures was two. All experiments were performed 24 h after transfection unless otherwise indicated.

### 4.3. Determination of the Amount of Pannexins mRNA

Total RNA from HDFs were isolated using the SV Total RNA Isolation System following the manufacturer’s instructions (Promega, Madison, WI, USA), and cDNA synthesis was performed using the SUPERSCRIPT III First-Strand System kit (Thermo Fisher Scientific, Waltham, MA, USA). Hot start PCR was performed using 2 ng cDNA in a total volume of 50 μL containing DreamTaq Green PCR Master Mix (Thermo Fisher Scientific, Waltham, MA, USA).

The primer sequences were as follows: human-Panx1, Forward: (5′-GCTATTACTTCAGCCTCTCC-3′), Reverse: (5′-CAGTATCTCCACCAAGAACC-3′); human-Panx3, Forward: (5′-AGGGCTGCTAAGTGATGAGA-3′), Reverse: (5′-GAGGTGTTTGGGTTTTGAGG-3′). PCR reactions were performed using 40 cycles and amplified PCR products were visualized in a 3% agarose gel.

### 4.4. Determination of Relative Amount of Proteins

The total amount of proteins was determined in cultured dermal fibroblasts and for siRNA-transfected dermal fibroblasts. For this, cells were lysed and homogenized in lysis buffer (150 mM NaCl, 10 mM Tris-HCl, pH 7.4, EDTA 2 mM, 1% Triton X-100, and 0.1% SDS), supplemented with a protease and phosphatase inhibitor cocktail (Thermo Scientific, Rockford, IL, USA). Protein concentration was then determined with the Qubit^®^ Protein Assay Kit (Thermo Scientific, Rockford, IL, USA). For Western analyses, 40 µg of protein was separated by 10% SDS-PAGE, followed by immunoblotting on PVDF membranes (BioRad, Hercules, CA, USA) with PBS containing 5% bovine serum albumin and 1% Tween-20 for 1 h at room temperature. Membranes were then incubated with specific antibodies against Panx1 (rabbit anti-Panx1, ABN242 Merck; 1:1000), Panx3 (rabbit anti-Panx3, 433,270 ThermoFisher, 1:1000), P2X7R (rabbit anti-P2X7, ab3458 Abcam; 1:1000), P2Y2 (mouse anti-P2Y2, ab168535 Abcam; 1:1000), PCNA (mouse anti-PCNA, P8825 Sigma-Aldrich, 1:1000), β-Tubulin (mouse anti-β-tubulin, sc-5274, Santa Cruz; 1:1000), and GAPDH (mouse anti-GAPDH, sc-47724, Santa Cruz; 1:1000). Membranes were then incubated with a horseradish peroxidase-labeled secondary antibody (HRP-linked secondary anti-mouse or anti-rabbit antibody (111-005-003, 315-005-003; Jackson Immuno Research; 1:5000) for 1 h at room temperature.

Finally, protein bands were visualized using an enhanced chemiluminescence kit (ECL, BioRad, Hercules, CA, USA) and detected using the image acquisition system Epichemi3 Darkroom (UVP Bioimaging System, Upland, CA, USA). Densitometric analysis was performed using the Image J software (version 1.49v; NIH, Bethesda, MD, USA).

### 4.5. In Vitro Scratch Wound Assay

Fibroblast migration was analyzed as previously described [74]. A wound was created through the cell monolayer using a 200 µL pipette tip. When required, cultured cells were then incubated with Panx1 channel blockers: 200 µM probenecid (PBN) (Thermo Fisher Scientific, Waltham, MA, USA), 100 µM ^10^Panx mimetic peptide, or scrambled mimetic peptide ^10^Panx (Sc ^10^Panx) 100 µM (Tocris, St. Louis, MO, USA). The wound areas were measured for 24 h, and standardized digital images were acquired every 2 h using a phase-contrast Nikon Eclipse TE200-U microscope (Nikon Instruments, Melville, NY, USA). Image analysis was performed using ImageJ (version 1.49v; NIH, MD, USA). Wound area  =  100% − percentage of the initial wound area size.

The purinergic receptor antagonists used here were: 10 µM A-740003 (Tocris, MO, USA), 100 µM suramin (Sigma-Aldrich, St. Louis, MO, USA), 200 µM Brilliant Blue G (BBG, Sigma-Aldrich, Louis, MO, USA), 10 µM AR-C118925XX (Tocris, St. Louis, MO, USA), 5 µM MRS-2768 (Tocris, Louis, MO, USA), and 1 mM ATP (Sigma-Aldrich, Louis, MO, USA). Purinergic receptor antagonists and agonists were incubated for 30 min before inducing the wound and maintained for the experiment’s duration.

### 4.6. Knockdown of Panx1 and Panx3 Expression by siRNA

The knockdown expression of either Panx1 and Panx3 was achieved using the psiRNA™-GFP System (InvivoGen, San Diego, CA, USA), with target sequences for Panx1 (GTGGACAAGATGGTCACATGT) and Panx3 (GTGAGCTGGACAAGTCTTACA). Fibroblast cultures growing in 35 mm plates at 60–70% confluence were transiently transfected with 2 µg siRNA using Lipofectamine 2000 (Thermo Fisher Scientific, Waltham, MA, USA) according to the manufacturer’s instructions. The GFP reporter gene was used to identify transfected cells. Digital images were acquired using a phase-contrast Nikon Eclipse T3E200-U microscope (Nikon Instruments, Melville, NY, USA) with a 20X objective. Images acquisition was every 3 h. The migration index used is defined as the proportion of cells expressing siRNA-GFP to non-transfected cells present in the migration front of the wound area (ten to fifteen cell rows from the migration front). Specific silencing was confirmed by Western blot.

### 4.7. Measurement of ATP Concentrations

Extracellular ATP was quantified using the luciferin-luciferase ATP determination assay kit (Thermo Fisher Scientific, Waltham, MA, USA). Fibroblasts were plated at 10,000 cells per well. Twenty-four hours later, 200 µM PBN or 10 µM A-740003 was added to different plates. After inducing the wound, the cell media were collected every 3 h and clarified by centrifugation at 1200× rpm. Five aliquots of 5 µL samples were collected after 30 min of incubation with the blocker and mixed with 45 µL ATP-mix solution in a white-bottomed 96-well plate (Corning Inc., Corning, NY, USA). Luminescence was determined using a BioTek Synergy HT plate reader (BioTek, Winooski, VT, USA). ATP levels were calculated from raw luminescence values, using the standard curves made with ATP concentration ranging from 1 nM to 1 µM in each inhibitor’s presence at the appropriate concentration.

### 4.8. Dye Uptake Assays

HDF cultures were initiated on 12 mm coverslips at different times in order to start wound healing assays every 3 h. HDFs were bathed in physiological extracellular solution containing 5 µM of ethidium bromide (Etd) and subjected to dye uptake time-lapse imaging as previously described [30]. Dye uptake was recorded using a Nikon Eclipse microscope (Nikon Instruments, Melville, NY, USA) with a 20X objective for time-lapse imaging. Images were captured using a CMOS digital camera (Hamamatsu, ORCA Flash 2.8, C11440, Hamamatsu, JP) every 30 s for 20 min to detect changes in fluorescence intensity. The mean Etd fluorescence intensity was measured in each ROI, after background subtraction. Background was evaluated on at least five regions devoid of cell bodies for each analyzed frame. For some experimental conditions, cultured HDFs were preincubated with 200 µM PBN, 100 µM ^10^Panx, or 100 µM Sc ^10^Panx for 15 min before and during the Etd uptake.

### 4.9. Single-Fibroblasts Motility Assay

Fibroblasts were plated on 25-mm glass coverslips at low confluence (20–30%) and maintained in recording solution for 1 h at room temperature. Single-cell images were acquired every 5 min as described above. One hour after the start of recording, the bath solution was replaced with vehicle or PBN 200 µM. The motility index was defined as the change in the fibroblast´s area (A_F_) at a fixed focal point for each recording time (t_n_) by the formula
$$A_{F}\;=\;\frac{A_{\mathrm{Ft}1}-A_{\mathrm{Ft}2}}{A_{\mathrm{Ft}2}}\;\times \;100$$

Cell borders were analyzed using ImageJ (NIH, Bethesda, MD, USA). Single-fibroblast motility was recorded for 40 min in control media (vehicle, VEH) and for another 200 min in media containing 200 µM PBN. Videos were recorded using a Nikon TE-2000U inverted microscope (Nikon ACT-2U, Melville, NY, USA), equipped with a 40X objective and DS-2WBc fast-cooled monochromatic digital camera.

### 4.10. Cell Surface Imaging of Actin Fibers

Fibroblasts growing at 70% confluences on 25-mm glass coverslips were transfected with 1 µg LifeAct-RFP using Lipofectamine 2000 (Thermo Fisher Scientific, Waltham, MA, USA). Single fibroblasts were recorded for one hour in control media (vehicle) and for another two hours in media containing 200 µM PBN. Images of surface actin were acquired every 3 min for 3 h using a Nikon Eclipse Ti-TIRF microscope (Nikon Instruments, Melville, NY, USA), equipped with a 60X oil TIRF objective (NA 1.49), laser line 488 and 532 nm, and CMOS digital camera (Hamamatsu, ORCA Flash 2.8, C11440, Hamamatsu, Japan).

### 4.11. Data Analysis and Statistics

For each data set, results are expressed as mean ± standard error (SEM); *n* refers to the number of independent experiments. Each set of experiments was conducted with experimental replicas. In most experiments, we used 6–10 experimental replicas from at least 6 different cell cultures. In the case of HDFs, the cell cultures were derived from 4 different human donors. Statistical analyses were performed using GraphPad Prism (version 6, Graphpad Software, Inc., San Diego, CA, USA). Detailed statistical results are included in the figure legends. Normality and equal variances of raw data were probed by the Shapiro–Wilk normality test. Unless otherwise stated, results were analyzed using unpaired two-tailed Student’s *t*-tests to compare two groups and ANOVA followed by Tukey’s or Bonferroni’s post hoc test in the case of multiple groups. A probability of *p* < 0.05 was considered statistically significant.

## 5. Conclusions

Our study suggests that Panx1 channels and P2X7Rs modulate the migration of human dermal fibroblasts. However, further investigation is required to identify the mechanism that modulates actin cytoskeleton dynamics under these conditions. Our findings suggest that Panx1 channels play an important role in controlling cell migration during wound healing and tissue regeneration.

## Figures and Tables

**Figure 1 ijms-22-01069-f001:**
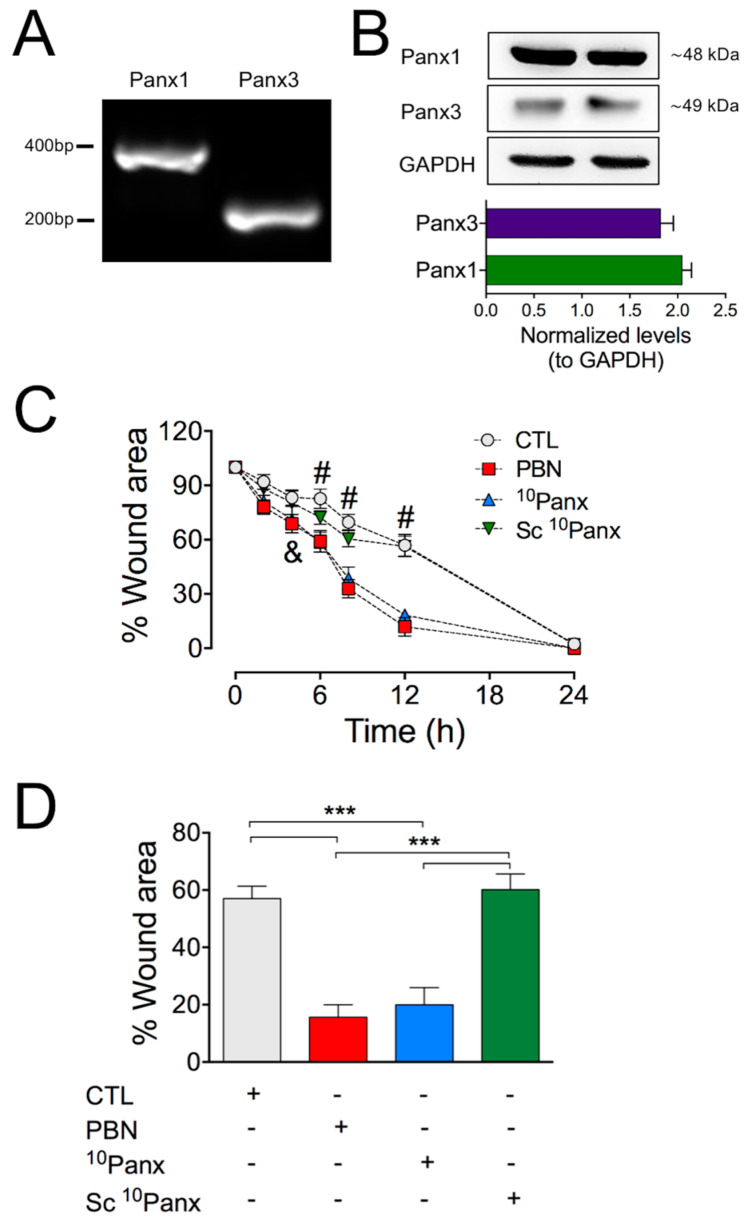
Inhibition of Panx1 channels accelerates wound closure in HDF cultures. (**A**) RT-PCR analysis of Panx1 and Panx3 mRNA in HDFs. (**B**) Representative Western blot showing protein levels of Panx1 and Panx3 in homogenates of HDFs. GAPDH immunodetection was used as a loading control. The graph shows the normalized levels of Panx1 and Panx3 in respect to GAPDH expression from 6 HDF cultures. (**C**) Time course of the wound healing process in HDF cultures. The scratch area was expressed as the percentage of wound area at each time relative to 0 h. CTL: control (untreated cultures). Panx1 channels were blocked using probenecid (PBN, 200 µM) and the mimetic peptide ^10^Panx1 (^10^Panx, 100 µM). The scrambled peptide was used as control (Sc ^10^Panx, 100 µM). Statistical significance was found between controls and PBN or ^10^Panx1. & = * *p* < 0.05 for 4 h; # = *** *p* < 0.001 for 6, 8, and 12 h; *n* = 10 (two-way ANOVA followed by Bonferroni’s post hoc test). (**D**) Comparison between wound healing rates 12 h after scratching for each treatment. *** *p* < 0.001; *n* = 10 (two-way ANOVA followed by Tukey’s post hoc test). All data are shown as mean ± SEM, *n* = 10 experimental replicas from at least four different cultures. Appendix A shows representative images of these experiments.

**Figure 2 ijms-22-01069-f002:**
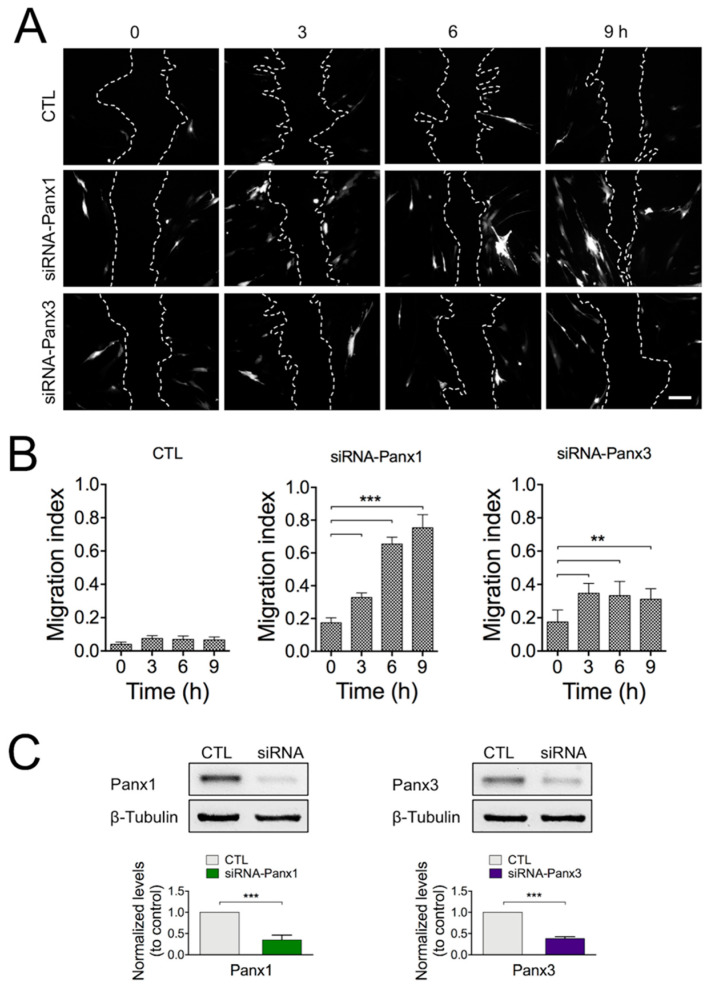
siRNA against Panx1 increases the migration index in HDF cultures. (**A**) Representative images of the wound healing process in cultures using Panx1 or Panx3 knockdown cells. HDFs transfected with siRNAs against Panx1, Panx3, or the control plasmid (CTL; siRNA-Empty-GFP) were identified by GFP expression. Dashed lines indicate the wound area (or the migration front line). Bar: 200 µM. Appendix A shows these same fluorescent images superimposed with their respective bright-field images. (**B**) Quantitative analysis of the average migration index 24 h after transfection of HDFs with respective siRNAs and CTL plasmid. Statistical significance ** *p* = 0.011; *** *p* < 0.001; *n* = 10 experimental replicas from 4 different cultures (one-way ANOVA followed by Tukey’s post hoc test). (**C**) Representative Western blots of Panx1 and Panx3 expression levels in total lysates from HDFs 12 h post-transfection. Graphs depicting the relative amount of Panx1 and Panx3 normalized to β-Tubulin from 4 different cultures. *** *p* < 0.001 for siRNA fibroblasts compared to control (unpaired two-tailed *t*-test). All data are shown as mean ± SEM.

**Figure 3 ijms-22-01069-f003:**
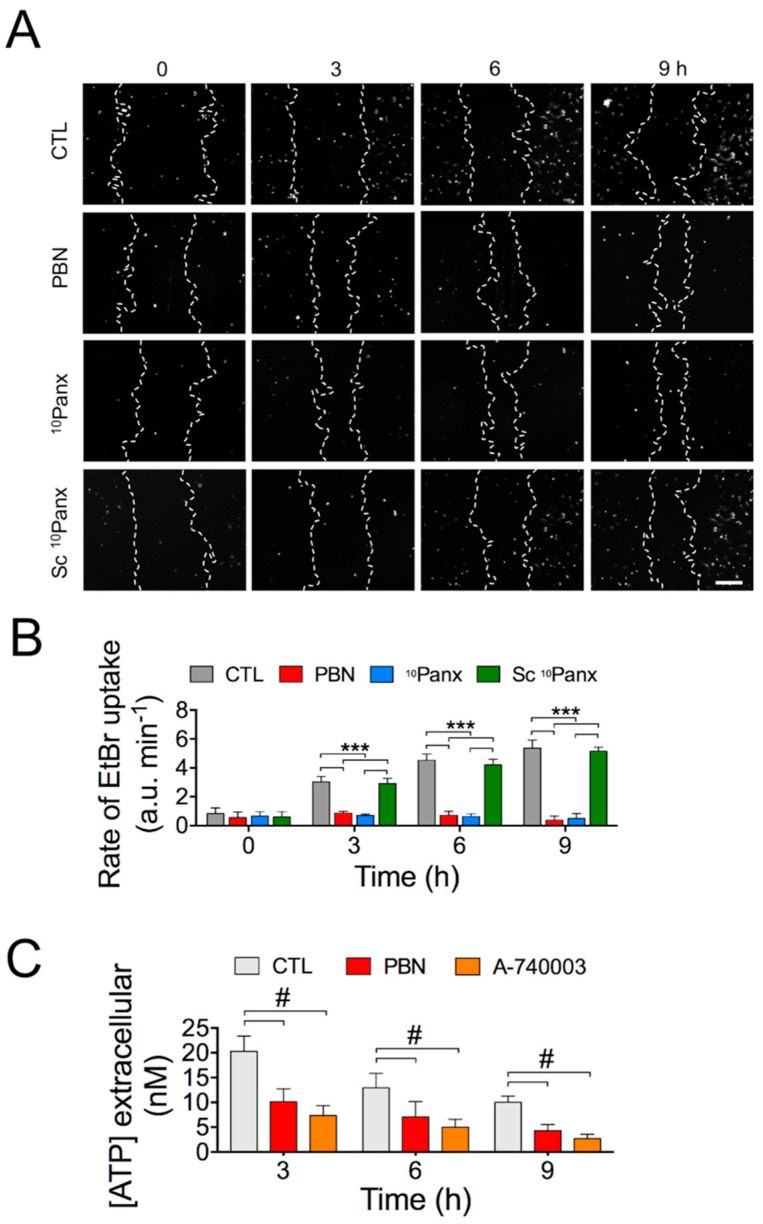
Activity of Panx1 channels during wound closure in HDFs. (**A**) Representative fluorescent images of Ethidium (Etd) uptake by HDF cells treated with 200 µM PBN, 100 µM ^10^Panx, 100 µM Sc ^10^Panx, or vehicle (Control, CTL). Bar: 200 µm. Appendix A shows these same fluorescent images superimposed with their respective bright-field image. (**B**) Etd uptake rate measure as described in Section 4; *** *p* < 0.001; *n* = 10 (two-way ANOVA followed by Bonferroni’s post hoc test). (**C**) Measurements of extracellular ATP released from HDFs as a function of time in the presence or absence of 200 µM PBN or 10 µM A-740003. #: *** *p* < 0.001; *n* = 10 (two-way ANOVA followed by Tukey’s post hoc test). All data are shown as mean ± SEM.

**Figure 4 ijms-22-01069-f004:**
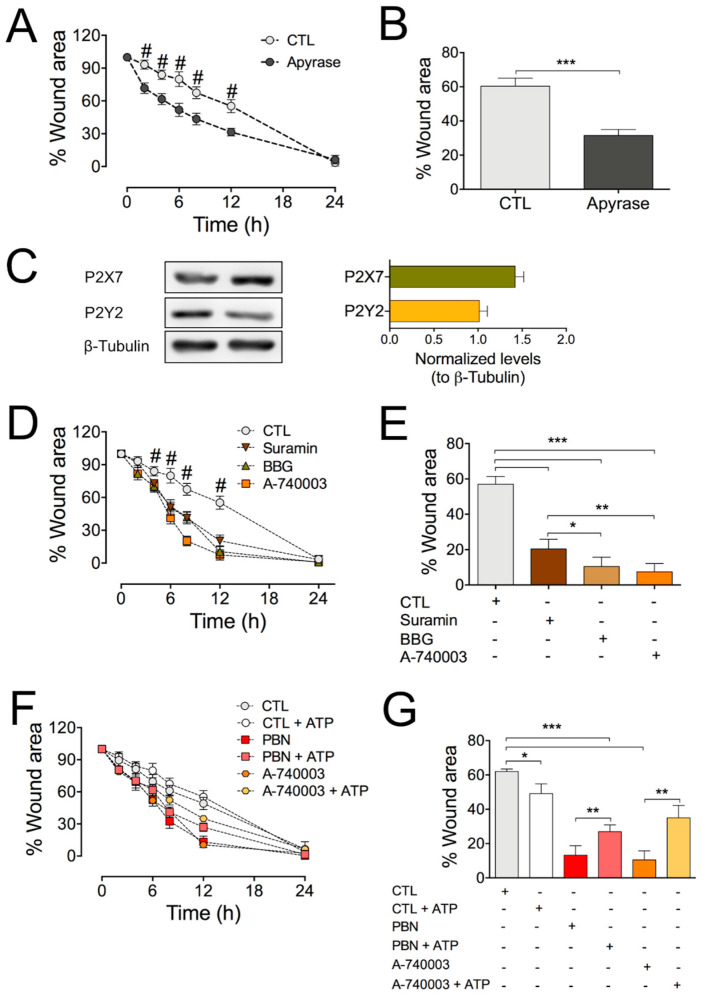
Blockade of the P2X7 receptor accelerates *in vitro* wound healing in HDF cultures. (**A**) Determination of the effect of exposure to 10 U/mL apyrase on the rate of wound healing assay in HDFs. Comparison between Apyrase and control (CTL); # = *** *p* < 0.001 for 2, 4, 6, 8, and 12 h; *n* = 8 (one-way ANOVA followed by Bonferroni’s post hoc test). (**B**) Comparison of the reduction in the wound area induced by respective treatments at 12 h of inducing the wound; *** *p* < 0.001; *n* = 8 (unpaired two-tailed t-test). (**C**) Representative Western blot images showing the expression of P2X7R and P2Y2R in two homogenates of HDF cells; β-tubulin was used as a loading control. Graphs depicting the relative amount of P2X7R and P2Y2R normalized to β-Tubulin from 8 different cultures. (**D**) Effects of purinergic receptor antagonists on wound healing assay in HDF cultures. Treatments against control (CTL). # = *** *p* < 0.001 for 6, 8, and 12 h; *n* = 10 (one-way ANOVA followed by Bonferroni’s post hoc test). (**E**) Percentage of wound area closure 12 h after inducing wound in HDF cultures treated with different purinergic receptor antagonists. (**F**) Determination of the wound healing rate under sustained application of 1 mM ATP in the absence or presence of PBN or A-740003 in HDFs. (**G**) Effect of exogenous ATP on wound area closure 12 h after inducing wound in absence or presence of PBN or A-740003. * *p* = 0.0260; ** *p* = 0.087; *** *p* < 0.001; *n* = 8 (one-way ANOVA followed by Bonferroni’s post hoc test). All data are shown as mean ± SEM.

**Figure 5 ijms-22-01069-f005:**
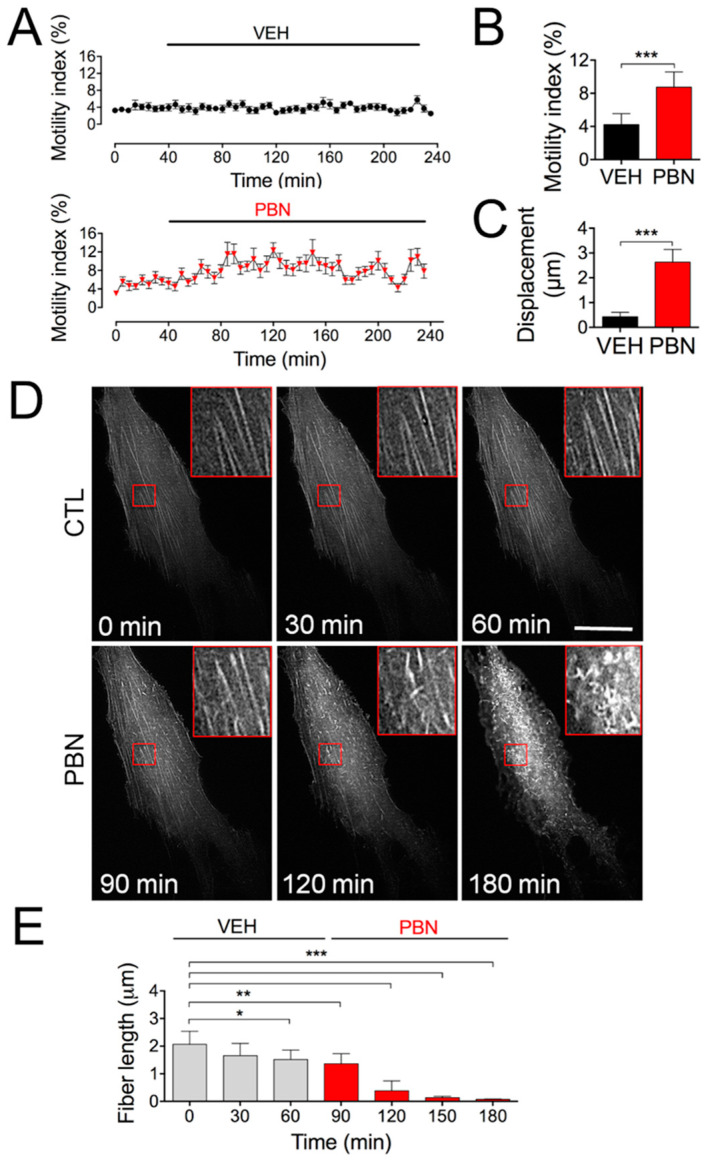
The blockade of Panx1 channels increases single-cell motility and the cell surface actin dynamics. (**A**) Quantification of motility index during treatment with vehicle (Control, VEH) or probenecid (PBN) in 20 individual HDFs per condition. (**B**) Quantification of the total average motility index from 20 independent cells analyzed for each condition. *** *p* = 0.006; (unpaired two-tailed Student’s *t*-tests); *n* = 10 cells from 4 different cultures. (**C**) Quantification of total single-cell displacements after 2 h of treatment with VEH or PBN. *** *p* < 0.001; *n* = 10 cells from 4 different cultures (unpaired two-tailed Student’s *t*-tests). (**D**) Representative in vivo imaging taken with TIRF microscopy of actin cytoskeleton dynamics in HDFs. The stress fibers were identified as Life-Act-RFP-labeled filaments; 200 µM PBN was added to the culture at 70 min; hence, the image taken at 120 min corresponds to 50 min treatment with PBN. The images on the right side are magnified images of the lower’s magnification, corresponding to red boxed areas. Bar: 25 µm. (**E**) Quantitative analysis of stress fibers’ length before and after incubation with PBN. * *p* = 0.02; ** *p* < 0.01; *** *p* < 0.001; *n* = 10 experiments (30 cells analyzed for each repeat) (one-way ANOVA followed by Bonferroni’s post hoc test). All data are shown as mean ± SEM.

**Figure 6 ijms-22-01069-f006:**
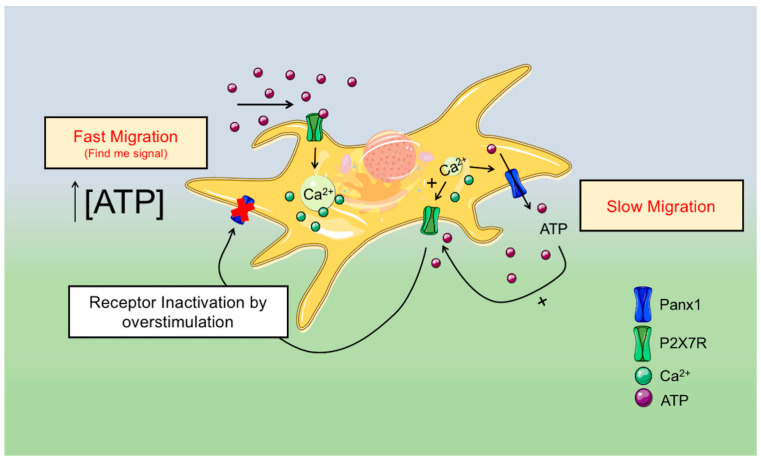
Proposed model for Panx1 channels’ role in HDF migration and cell surface actin dynamics.

## Data Availability

The data presented in this study are available on request to the corresponding author.

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
