# Peer review of "Restraint of Human Skin Fibroblast Motility, Migration, and Cell Surface Actin Dynamics, by Pannexin 1 and P2X7 Receptor Signaling"

_ijms, 2021, doi:10.3390/ijms22031069_

Round 1

Reviewer 1 Report

Hi

Thank you for provide a good manuscript with provocative results and future questions in the area. Overall the main issue of the manuscript is the description of the experiments and results as wel as the English. Some examples (No all are included)

Line 31, ... healing process induced by....

line 33, mice fibroblast required for a migratory phenotype 

line 33, eliminate further

line 33, release of ATP by a pannexin-1 dependent mechanism

In the entire manuscript, replace activity by opening 

line 41, Are any non-pore channels ????

line 44, lading replaced by leading 

Line 45, coupling replaced by interactions 

Line 48, activating replaced by suggesting 

line 53 replace for normal -- during normal

line 62, ...this cell type in response to 

These are some examples. 

In the results section

  • Why Panx2 was not analyzed?
  • IN Fig. 1C, are all the numbers significant?
  • In all the figures, of wound healing, the word I think is delay, because the end point is always the same 
  • Some stats need clarification, *, ** and ** some figures does not have
  • The wound healing figures such as Fig. 2A and 3A, could benefit a lot of a corresponding phase. 
  • The GFP staining is poor in all the conditions. Is any explanation for that?
  • How long the siRNA reduces expression? 
  • line 138, what mean times of healing, healing is a process  
  • In Figure 4, the data is clear thus, terms unclear messages need to the removed 
  • The cell rearrangement need to be explained better after the 1 h of recording.
  • The use of tirf microscopy to determine uptake only limit the approach is really TIRF was used due that TIRF only see close to the glass
  • Are any carahtherization of the human fibroblast?
  • In the methods, the company invitrogen fused with thermo, more than 10 year ago. 
  • Nikon is from Japan, no NY

Author Response

Answer to reviewers:

rev 1:

Thank you for provide a good manuscript with provocative results and future questions in the area. Overall the main issue of the manuscript is the description of the experiments and results as wel as the English. Some examples (No all are included)

Line 31, ... healing process induced by....

line 33, mice fibroblast required for a migratory phenotype 

line 33, eliminate further

line 33, release of ATP by a pannexin-1 dependent mechanism

In the entire manuscript, replace activity by opening 

line 41, Are any non-pore channels ????

line 44, lading replaced by leading 

Line 45, coupling replaced by interactions 

Line 48, activating replaced by suggesting 

line 53 replace for normal -- during normal

line 62, ...this cell type in response to 

These are some examples. 

Response: We modify the text as suggested by reviewers. In addition, the manuscript was reviewed by a colleague (John Ewer) who is an English native speaker. All corrections and additions are labeled using the word processor track changes.

In the results section

  • Why Panx2 was not analyzed?. Response: We did not incorporate in this study Panx2, because its expression outside of the nervous system is a matter of debate (Baranova et al., 2004; Genomics 83, 706–716. doi: 10.1016/j.ygeno.2003.09.025. Bruzzone et al., 2003; Proc. Natl. Acad. Sci. U S A 100, 13644–13649. doi: 10.1073/pnas.2233464100. Le Vasseur et al., 2014; Front Cell Neurosci 8:392. doi: 10.3389/fncel.2014.00392). In addition, expression of Panx2 is mostly intracellular (Boassa et al., 2015; Front Cell Neurosci 2;8:468. doi: 10.3389/fncel.2014.00468. Le Vasseur M, et al., 2019; Cancers (Basel) 11(3):343. doi: 10.3390/cancers11030343), therefore, it seems that it does not have a role as a plasma membrane channel.
  • IN Fig. 1C, are all the numbers significant? Response: We incorporate more statistic data in this and other figures.
  • In all the figures, of wound healing, the word I think is delay, because the end point is always the same. Response: I think this may be a misunderstanding of reviewer. In fact, the treatment with PBN or mimetic peptide 10Panx1 accelerates the wound healing process; these treatments produce maximum closure of the healing almost 12 h before it does it in control cultures.
  • Some stats need clarification, *, ** and ** some figures does not have: Response: In figure legends we incorporate the P values for the respective symbols.
  • The wound healing figures such as Fig. 2A and 3A, could benefit a lot of a corresponding phase. Response: We incorporate the bright-field images in the supplementary figures S1 A-B and S3A.
  • The GFP staining is poor in all the conditions. Is any explanation for that? Response: These are transient transfections, therefore part of the cells were not transfected and they were use as internal control for the experiments. We analyzed the number of cells expressing the siRNA-GFP relative to non-transfected cells in the migration front of the wound area. We incorporate new representative images for Figure 2A and the corresponding bright field images in supplementary figure S3.
  •  
  •  
  • How long the siRNA reduces expression? Response: Unfortunately, we cannot answer this question. We have done only one point for the determination of effectiveness of siRNA treatments, which was 12 h after the end of transfection protocol, that is 24 hrs after the addition of the plasmid.
  • line 138, what mean times of healing, healing is a process  Response: The reviewer is right, we modify this in the text.
  • In Figure 4, the data is clear thus, terms unclear messages need to the removed Response: We did not understand this question.

The cell rearrangement need to be explained better after the 1 h of recording. Response: We incorporate a better explanation in figure legend: “The stress fibers were identified as Life-Act-RFP labeled filaments; 200 µM PBN was added to the culture at 70 min, hence the images taken at 120 min correspond to 50 min treatment with PBN”.

  • The use of tirf microscopy to determine uptake only limit the approach is really TIRF was used due that TIRF only see close to the glass… Response: We thanks the reviewer for this coment. We used conventional transmission fluorescent microscopy for dye uptake assays, and TIRF was used only for actin filaments rearrangements studies. There was a mistake in method section that we have corrected.
  • Are any carahtherization of the human fibroblast?

Response: We incorporate this relevant information on the Method section.

Primary human dermal fibroblasts (HDF) were provided by Dr. Caroline Weinstein. HDF were obtained from biopsies of healthy abdominal skin from 4 patients who underwent regenerative autograft surgery. Only remnant, clinically healthy skin, not required for diagnostic or surgery purposes was used. Cell culture from biopsies were prepared as in (C.R. Weinstein-Oppenheimer et al. / Materials Science and Engineering C 79 (2017) 821–830) with modification. First, the biopsy was extensively washed with sterile phosphate saline buffer (pH 7.35). Then it was mechanically disrupted using sterile scalpels and incubated in collagenase (3 mg/mL) (Nordmark, Germany; medical grade) for 1 h within an incubator (37 °C, 5% CO2 and humidified atmosphere). After this, the mixture was centrifuged at 553g, and the pellet was cultured in Dulbecco’s modified Eagle’s medium: nutrient mixture F-12 (Thermo Fisher Scientific, MA, USA), 10% inactivated fetal bovine serum (Biological Industries, Israe) 10,000 IU/mL penicillin and 10 mg/ml streptomycin (Thermo Scientific Inc. ®) in a 5% CO2 atmosphere at 37°C.

Unpublished information: As we said HDF cultures were from Dr. Weinstein´s laboratory. Dr. Weinstein leads a group of Chilean researchers that have develop hybrid biomaterial to be used as a wound dressing. They are improving the effectiveness by integration of an autologous clot hydrogel carrying mesenchymal stem cells onto the biopolymeric scaffold (Weinstein-Oppenheimer et al., 2017; Materials Science and Engineering C 79 (2017) 821–830). In those studies, Dr. Weinstein used cells from human gingiva tissue and foreskin, and not from abdominal skin, which has not been published yet because and is part of another study that is under development. HDF from abdominal skin has been characterized by morphological criteria, flow cytometry, TGFß-3 secretion, and differentiation towards bone, cartilage, and adipocyte lineage. These unpublished results showed that:

  1. These cultures presented long spindle-shaped cells, stellate-shaped cells with fibroblastic extensions, and small round dividing primary cells
  2. Produce 2 ng/mL TGFß-3 after three days in conventional 1D cell culture.
  3. More than 90% express CD44 and CD105 and show no expression of CD34 or CD45.
  4. Exhibit the capability to differentiate to the three mesenchymal lineages bone, cartilage and adipocyte. In summary, the human dermal fibroblast exhibits feature of mesenchymal stem cells.
  • In the methods, the company invitrogen fused with thermo, more than 10 year ago.  Response: Yes, that right we change this.
  • Nikon is from Japan, no NY, Response: Yes, that right we modify this.

Reviewer 2 Report

The authors present an interesting exploration of the role pannexin channels play in human fibroblast function.

Major revisions

1. Supplementary Data has not been included. Please provide for review.

Minor revisions

  1. Line 74 states "Pannexin expression in human-derived skin fibroblast (Hu-Fb) has not been reported yet." However, this has been reported elsewhere www.ncbi.nlm.nih.gov/pmc/articles/PMC3779754/ (Fig 4B) and https://pubmed.ncbi.nlm.nih.gov/24522432/. This statement is misleading and requires removal.
  2. Primary human dermal fibroblasts (HDF) is more conventional than human-derived skin fibroblast (Hu-Fb). Please replace throughout.
  3. What was the source of HDF? What anatomical location were they grown from? What method was used to extract and culture the HDF? Anatomical source can greatly impact fibroblast function and responses and is therefore relevant to specify www.sciencedirect.com/science/article/pii/S0022202X17330804. How many experimental replicates were performed? How many biological replicates were conducted?
  4. N number for Figure 1 are lacking from text and legend (were replicates experimental or biological ie different patients)?
  5. Figure 1 would benefit from representative scratch assay images which should clearly show the scratch and peripheral dermal fibroblasts.
  6. The transfection efficiency in Figure 2A looks lower than reported in Figure 2C. Representative bright field images for 2A should be shown alongside (showing all the fibroblasts, not just those transfected). Was the analysis conducted on bright field or GFP images?
  7. Figure 4 - n numbers are only provided for panel E.

Author Response

Rev2:

The authors present an interesting exploration of the role pannexin channels play in human fibroblast function.

Major revisions

  1. Supplementary Data has not been included. Please provide for review. Response: We did not know what happen because we uploaded it together with the main manuscript. We send it again with the modifications.

Minor revisions

  • Line 74 states "Pannexin expression in human-derived skin fibroblast (Hu-Fb) has not been reported yet." However, this has been reported elsewhere www.ncbi.nlm.nih.gov/pmc/articles/PMC3779754/ (Fig 4B) and https://pubmed.ncbi.nlm.nih.gov/24522432/. This statement is misleading and requires removal. Response: We thank the review for point this out, we made the modification in the text. “As previously described and characterized by [20], [22], we found using RT-PCR and Western blot analyses of HDF cultures that Panx1 and Panx3 are expressed in primary HDF.
  • Primary human dermal fibroblasts (HDF) is more conventional than human-derived skin fibroblast (Hu-Fb). Please replace throughout. Response: We made the substitution through the text and figures.
  • What was the source of HDF? What anatomical location were they grown from? What method was used to extract and culture the HDF? Anatomical source can greatly impact fibroblast function and responses and is therefore relevant to specify www.sciencedirect.com/science/article/pii/S0022202X17330804. How many experimental replicates were performed? How many biological replicates were conducted?...

Response: We incorporate this relevant information on the Method section.

Primary human dermal fibroblasts (HDF) were provided by Dr. Caroline Weinstein. HDF were obtained from biopsies of healthy abdominal skin from 4 patients who underwent regenerative autograft surgery. Only remnant, clinically healthy skin, not required for diagnostic or surgery purposes was used. Cell culture from biopsies were prepared as in (C.R. Weinstein-Oppenheimer et al. / Materials Science and Engineering C 79 (2017) 821–830) with modification. First, the biopsy was extensively washed with sterile phosphate saline buffer (pH 7.35). Then it was mechanically disrupted using sterile scalpels and incubated in collagenase (3 mg/mL) (Nordmark, Germany; medical grade) for 1 h within an incubator (37 °C, 5% CO2 and humidified atmosphere). After this, the mixture was centrifuged at 553g, and the pellet was cultured in Dulbecco’s modified Eagle’s medium: nutrient mixture F-12 (Thermo Fisher Scientific, MA, USA), 10% inactivated fetal bovine serum (Biological Industries, Israe) 10,000 IU/mL penicillin and 10 mg/ml streptomycin (Thermo Scientific Inc. ®) in a 5% CO2 atmosphere at 37°C. 

Unpublished information: As we said HDF cultures were from Dr. Weinstein´s laboratory. Dr. Weinstein leads a group of Chilean researchers that have develop hybrid biomaterial to be used as a wound dressing. They are improving the effectiveness by integration of an autologous clot hydrogel carrying mesenchymal stem cells onto the biopolymeric scaffold (Weinstein-Oppenheimer et al., 2017; Materials Science and Engineering C 79 (2017) 821–830). In those studies, Dr. Weinstein used cells from human gingiva tissue and foreskin, and not from abdominal skin, which has not been published yet because and is part of another study that is under development. HDF from abdominal skin has been characterized by morphological criteria, flow cytometry, TGFß-3 secretion, and differentiation towards bone, cartilage, and adipocyte lineage. These unpublished results showed that:

  1. These cultures presented long spindle-shaped cells, stellate-shaped cells with fibroblastic extensions, and small round dividing primary cells
  2. Produce 2 ng/mL TGFß-3 after three days in conventional 1D cell culture.
  3. More than 90% express CD44 and CD105 and show no expression of CD34 or CD45.
  4. Exhibit the capability to differentiate to the three mesenchymal lineages bone, cartilage and adipocyte. In summary, the human dermal fibroblast exhibits feature of mesenchymal stem cells.

N number for Figure 1 are lacking from text and legend (were replicates experimental or biological ie different patients)?. Response: Each set of experiments is conducted with 10 experimental replicas (covers) from at least 4 different cultures; the cultures came from different human (donor) or mice. We incorporate in methods a paragraph explaining this (data analysis and statistic).

  • Figure 1 would benefit from representative scratch assay images which should clearly show the scratch and peripheral dermal fibroblasts. Response: We incorporate the images in the supplementary Figure S1.
  •  
  • The transfection efficiency in Figure 2A looks lower than reported in Figure 2C. Representative bright field images for 2A should be shown alongside (showing all the fibroblasts, not just those transfected). Was the analysis conducted on bright field or GFP images?. Response:

These are transient transfections, therefore part of the cells were not transfected and they were use as internal control for the experiments. We analyzed the number of cells expressing the siRNA-GFP respect to non-transfected cells in the migration front of the wound area through the time during different treatments. We incorporate new representative images for Figure 2A and the corresponding bright field images in supplementary figure S3.

  • Figure 4 - n numbers are only provided for panel E. Response: we incorporate the n for this and other experiments that were missing, thank you.

Round 2

Reviewer 1 Report

The current version was well reorganized and described. Congratulations. 

After read the manuscript I have only 1 question 

Why A740003 blocks ATP secretion? If the model in Fig. 6 is correct, the P2X7 blocker only suppuost to prevent the receptor activation. May be ATP is also secreted by the P2X7 receptors. Please add a section explaining this result 

Author Response

Hello, thank you very much for asking this question.  We incorporated a sentence about this point in the discussion section (we incorporate a couple of references to support the arguments) that is labeled in yellow in the main text.

"The latter is consistent with our finding that the P2X7Rs blocker A-740003 is capable of inhibiting ATP release from HDFs and MDFs, but not from MDFpanx1-/-, suggesting that in our model, ATP may also activate Panx1 channels, supporting the idea that the activation of P2X7R could be both upstream and downstream of Panx1 channel activation. The latter agrees with previous findings that P2X7R activation causes ATP release [50], [51] and activation of Panx1 channels [7], [8], [52]. However, we cannot discard that the inhibition of P2X7R with A-740003 affects the Panx1-independent mechanism of ATP release. For example, it has been proposed that the P2X7R itself can form a non-selective plasma membrane channel (or macropore) that is permeable to large hydrophilic molecules, like dyes or ATP [53], [54]".  

Reviewer 2 Report

The updated manuscript has adequately addressed the Reviewer's comments. No further comments.

Author Response

Thank You very much